# Risk Factors of Urothelial Cancer in Inflammatory Bowel Disease

**DOI:** 10.3390/jcm10153257

**Published:** 2021-07-23

**Authors:** Gian Paolo Caviglia, Giorgio Martini, Angelo Armandi, Chiara Rosso, Marta Vernero, Elisabetta Bugianesi, Marco Astegiano, Giorgio Maria Saracco, Davide Giuseppe Ribaldone

**Affiliations:** 1Department of Medical Sciences, Division of Gastroenterology, University of Torino, 10126 Torino, Italy; gianpaolo.caviglia@unito.it (G.P.C.); giorgiomartini96@gmail.com (G.M.); chiara.rosso@unito.it (C.R.); elisabetta.bugianesi@unito.it (E.B.); giorgiomaria.saracco@unito.it (G.M.S.); 2Department of Internal Medicine, San Matteo Hospital, 27100 Pavia, Italy; martavernero@gmail.com; 3Department of General and Specialist Medicine, Gastroenterologia-U, Città della Salute e della Scienza di Torino, C.so Bramante 88, 10126 Turin, Italy; mastegiano@cittadellasalute.to.it

**Keywords:** Crohn’s disease, ulcerative colitis, malignant, neoplasm, urinary, bladder, ureter, urethra, urothelium

## Abstract

Extraintestinal cancers are important complications in patients with inflammatory bowel disease (IBD). A limited number of publications are available regarding the association between IBD and urothelial cancer. The primary outcome of our study was the comparison of the prevalence of urothelial cancer in patients with IBD with respect to the prevalence in the general population. Secondary outcomes were the assessment of risk factors for the onset of urothelial cancer in IBD. In a retrospective study we examined the medical records of all patients with a confirmed diagnosis of IBD followed in our clinic between 1978 and 2021. For each of the patients with identified urothelial cancer, more than ten patients without cancer were analyzed. Furthermore, 5739 patients with IBD were analyzed and 24 patients diagnosed with urothelial cancer were identified. The incidence of urothelial cancer, compared with the incidence in the general population, was not significantly different (0.42% vs. 0.42%; *p* = 0.98). Twenty-three cases were then compared (1 case was discarded due to lack of follow-up data) against 250 controls. During the multivariate analysis, smoking (odds ratio, OR = 8.15; 95% confidence interval, CI = 1.76–37.63; *p* = 0.007) and male sex (OR = 4.04; 95% CI = 1.29–12.66; *p* = 0.016) were found as risk factors. In conclusion, patients with IBD have a similar risk of developing urothelial cancer compared to the general population, but males with a history of smoking are at increased risk.

## 1. Introduction

Inflammatory bowel diseases (IBD) are a heterogeneous group of immune-mediated diseases of unknown etiology that can affect the digestive tract in a variable manner. Traditionally they are grouped into two entities, Crohn’s disease (CD) and ulcerative colitis (UC), whose differential diagnosis is based on clinical, histological, laboratory, and endoscopic data. A third entity, unclassified IBD (IBD-U), represents a temporary diagnosis until there are sufficient elements to define whether it is CD or UC [1].

Regarding risk factors, smoking has different effects on IBD. In CD, smoking worsens the course of the disease, leads to a reduced response to medical therapy, and an increased risk of exacerbations and complications, with a more aggressive disease profile and increased surgery rate. The role of smoking in the onset of the disease is unclear, but the higher incidence of CD among smokers would seem to suggest that smoking is part of the events underlying the pathogenesis of the disease; nicotine and its derivatives can directly influence the immune responses of the mucosa, the composition of the microbiome, the production of pro-inflammatory cytokines, the tone of smooth muscle, and intestinal permeability, acting at the vascular level causing coagulation [2]. The use of tobacco, on the other hand, seems to protect against UC and reduce its severity, although it does not seem to improve the natural history of the disease. On the contrary, ex-smokers have a higher risk of developing the disease, which, in these cases, is often more extensive and more refractory to therapy than the disease in individuals who have never smoked [3].

A link between IBD and extraintestinal cancers has been observed in people with IBD [4]. Though the mechanism behind IBD and oncogenesis is unknown, it has been observed that inflammation not only acts as the host’s response to malignant tumors, but also triggers carcinogenesis [5]. Furthermore, immunosuppressive medicines, which are commonly used to treat IBD, are thought to be linked to an increased risk of malignancies such as non-Hodgkin lymphoma, acute myeloid leukaemias, non-melanoma skin cancers, and urinary tract cancers [6].

Tumors of urothelial tissue are neoplasms that develop at the level of the transitional epithelium (the tissue that comes into contact with urine and covers the urinary tract from the renal calyxes to the urethra). The bladder is the most frequent site but tumors of the transitional epithelium may also be present in the following areas: renal calyxes, ureters, and urethra. Bladder cancer accounts for 3% of cancers diagnosed globally and is particularly prevalent in the Western world [7]. Tobacco is the best known of the factors favoring the development of bladder cancer, in particular active cigarette smoking is responsible respectively for 60% and 30% of all urothelial carcinomas in males and females [8].

Only a limited number of publications are available in the literature regarding the possible association between urothelial cancer and IBD with conflicting results [9,10], as well as regarding the possible role of azathioprine as a risk factor [11,12].

The aim of our study was to evaluate the frequency of urothelial cancer in a large series of patients with IBD and to search for possible risk factors, including medications.

## 2. Materials and Methods

We retrospectively reviewed the medical records of all patients with IBD followed in the gastroenterology clinic between 1978 and 2021, with the aim of assessing the frequency of urothelial cancer and comparing it with that of the general population of the same country. Patients were included in the “IBD cohort” (Ethics Committee Approval No. 0056924).

Inclusion criteria were:-all patients with a confirmed diagnosis of IBD according to the indications of the European Crohn’s and Colitis Organization (ECCO) [13];-minimum age of 16, with no upper age limits.

Exclusion criterion was:-lack of data on the presence of tumor comorbidities.

The primary outcome was the comparison of the frequency of urothelial cancer in our population with respect to the data available from the AIOM (Italian Association of Medical Oncology) guidelines for the general Italian population [14].

For each of the patients with identified urothelial cancer (cases), more than ten IBD patients who did not develop urothelial cancer (controls) were randomly selected (alphabetically) from the medical records of all patients with IBD followed in the gastroenterology clinic between 1978 and 2021.

The secondary outcomes were the assessment of risk factors for the onset of urothelial cancer in patients with IBD.

The risk factors assessed with a univariate analysis were:-categorical variables: sex, smoking habit, type of IBD, treatment with mesalazine, treatment with thiopurine, treatment with anti-tumor necrosis factor (TNF), treatment with anti-integrins, and surgical treatment for intestinal disease;-continuous variables: age at diagnosis of IBD, duration of treatment with mesalamine, duration of treatment with thiopurine, duration of treatment with anti-TNF, and duration of treatment with anti-integrins.

### Statistical Analysis

The normal distribution of continuous variables was assessed using the D’Agostino–Pearson test. Continuous variables not normally distributed were reported as median and interquartile range (IQR), continuous variables normally distributed were reported as mean ± standard deviation (SD). Categorical variables were reported as numbers and percentages. The Mann–Whitney test and the chi-square test were used to compare continuous and categorical variables, respectively. Multivariate analysis (logistic regression) was then carried out by inserting the variables of clinical interest. A *p*-value of less than 0.05 was considered statistically significant. All statistical analyses were performed using MedCalc v.18.9.1 (MedCalc Software Ltd., Ostend, Belgium).

## 3. Results

In the first phase, we analyzed 5739 patients followed in our clinic from 1978 to 2021, with a confirmed diagnosis of IBD and with available data on the presence of any comorbid tumors; of these, 24 received a diagnosis of urothelial neoplasia (0.42%). Specifically, 20 patients (83.3%) had a bladder tumor, 3 patients (12.5%) had a ureteral tumor, 1 patient (4.2%) had no precise location available, no patient had a tumor in the urethral or kidney pelvis. From the point of view of tumor typing, the information present was partially incomplete and some histological information regarding the tumor was missing. According to the data in our possession, from the point of view of tumor grading, 9 (23.1%) tumors were infiltrating (6 tumors were pTa, 3 were pT1), and 1 (4.4%) tumor was infiltrating (pT3). From the point of view of tumor staging, 2 (8.7%) tumors were G1, 7 (30.4%) tumors were G2, and 2 (8.7%) tumors were G3. Comparing the frequency of urothelial cancer in our patients affected by IBD with that of the whole Italian population, it was no different: 0.42% in both groups (*p* = 0.98).

One patient diagnosed with urothelial cancer was excluded from the second phase (case-control study) due to lack of follow-up data; 250 randomly selected control patients were analyzed. The mean age at diagnosis of urothelial cancer in the 23 patients was 61.3 ± 13.0 years. In 6 cases (26%) the diagnosis of urothelial cancer preceded the diagnosis of IBD (all active smokers), and in 17 cases (74%) it was subsequent. In the 17 patients in whom the diagnosis of urothelial cancer was subsequent to that of IBD, 8.0 ± 10.1 years elapsed between the diagnosis of IBD and the diagnosis of urothelial neoplasia.

The risk factors for developing urothelial cancer are shown in Table 1.

The frequency of urothelial cancer by gender is shown in Figure 1.

Figure 2 shows smoking habits in the 2 groups.

The age comparison at diagnosis of IBD in the 2 groups is shown in Figure 3.

No patient diagnosed with urothelial cancer received anti-integrin treatment.

In multivariate analysis, the factors that were statistically associated with the development of urothelial cancer in the patient with IBD were male sex (odds ratio, OR = 4.04; 95% confidence interval, CI = 1.29–12.66, *p* = 0.016) and a history of cigarette smoking (OR = 8.15, 95% CI = 1.76–37.63, *p* = 0.007). Diagnosis of CD (OR = 1.49, 95% CI = 0.53–4.19, *p* = 0.44), treatment with mesalamine (OR = 0.33, 95% CI = 0.056–1.93, *p* = 0.22), and treatment with thiopurines (OR = 0.57, 95% CI = 0.19–1.69, *p* = 0.40) were not risk factors.

## 4. Discussion

Malignant tumors, both gastrointestinal and extraintestinal, are known long-term complications in patients with IBD; in fact, the latter have a long-term risk of cancer that is 30% higher for gastrointestinal cancers and 10% for extraintestinal cancers compared to the general population. This could be a consequence of both chronic inflammation and the consequent important use of immunosuppressive drugs to control inflammation [15,16]. While colon cancer is likely to be associated with the inflammatory state caused by intestinal disease, regarding extraintestinal cancers the cause of the increased risk is not well-known. Studies have found a correlation between IBD and an increased risk of a limited number of extraintestinal cancers, such as hematological cancers and lung cancer [4,17,18]. Furthermore, patients with IBD appear to have a slightly higher risk of cancers from the prostate, skin, liver, and biliary system [19,20]. The underlying mechanisms are not clear since tumor etiology is more often multifactorial.

On the other hand, the possibility that chronic intestinal inflammation can be associated with tumors at the urothelial level is widely discussed in the literature. Studies on this are few and show different results. In particular, the study by Madanchi et al. [16] and the study by Algaba et al. [21] are single-center cohort studies and have shown a statistically significant increase in urothelial cancer in patients with IBD. However, the first study evaluates 1026 patients, while the second 590, thus introducing a possible bias linked to the small size of the selected patient cohort. In fact, these results seem isolated, while there are more publications that do not find any association between the two diseases in question [10,15,17,22,23,24,25,26,27].

In our study, comparing the frequency of urothelial cancer in the population with IBD with that of the Italian population, we observed that a diagnosis of IBD is not a risk factor for the development of urothelial neoplasm (*p* = 0.98). This data takes on an important value since the relevant sample of 5739 patients on which the analysis was carried out makes our study the largest performed on a population of patients belonging to the same center. However, a statistically significant increased risk of tumor was found in patients diagnosed with IBD in men (OR = 4.04; 95% CI = 1.29–12.66) and in patients with a history of cigarette smoking (OR = 8.15; 95% CI = 1.76–37.63). Regarding the risk of urothelial cancer in association with smoking, the study by Algaba et al. [21] found an increased risk of urothelial cancer in smoking IBD patients, while no association was found in the study by Madachi et al. [16].

Regarding the risk of urothelial cancer in association with CD, the study by Kappelman et al. [15] found no increased risk (standardized incidence ratio, SIR = 1.2, 95% CI = 0.8–1.6). Conversely, in the study by Pedersen et al., there was a significant association between CD and urothelial cancer (SIR = 2.03, 95% CI = 1.14–3.63) [4]. In our study, despite a higher prevalence of smoking history in the CD population, the diagnosis of urothelial cancer was not statistically significantly increased.

Important is the finding in our study of an increased frequency of urothelial cancer in male patients with IBD. This figure is in line with that of the general population: an increased incidence of urothelial cancer in men is known (24,000 new diagnoses in men against 5700 diagnoses in women, in 2019) [7]. This is mainly explained by the different incidence of risk factors in the male population, mainly with regard to smoking and occupational exposure to carcinogens [8,28].

Some studies also reported that long-term use of immunosuppressive drugs, particularly thiopurines, in patients with IBD may increase the risk of cancer [4,15,29]. Immunosuppressive drugs, which are widely used for IBD treatment, are thought to be responsible for a small increased risk of extraintestinal cancers such as non-Hodgkin’s lymphoma, non-melanoma skin, and urinary tract cancers [6,30,31]. In this regard, it is believed that, while the use of immunosuppressants can guarantee greater protection from tumors in the intestine, acting on chronic inflammation and reducing the risk of neoplastic transformation, the same effect may not be present for extraintestinal tumors, whose pathogenesis could be different from that of chronic inflammation. In contrast, the study by Pedersen et al. suggested a possible bias derived from increased clinical attention in patients undergoing treatment with immunosuppressants and a greater likelihood of being subjected to diagnostic investigations in these patients [4]. Regarding this debated topic, our study seems to deny a possible correlation. There was no significant association between thiopurine treatment and urothelial cancer (OR = 0.57; 95% CI = 0.19–1.69). Our study cannot draw any conclusions regarding a possible association between the use of anti-TNF or anti-integrins and the occurrence of urothelial tumors in patients with IBD due to the insufficient number of patients belonging to these groups. 

We want to underline data of particular interest, namely, the age at the diagnosis of IBD in patients in the urothelial cancer group. In these cases, it was found to be 56.0 years (IQR = 40.0–64.8). This was found to be definitely more advanced than the control group: 32.0 years (IQR = 22.0–44.0), much more in line with the epidemiological data regarding IBD [32]. This data is not easy to interpret, but we suggest the particular characteristics of the patients that develop urothelial cancer as a possible explanation: the patients in question are more frequently males, smokers, and workers in industrial activities, patients in whom a late diagnosis is perhaps more frequent for a reduced understanding of risk factors for health and late access to health care for symptom assessment. 

From the urological point of view, the results found are in line with the epidemiological characteristics of urothelial tumors in the general population. In our population, patients with bladder cancer were 83.3% and patients with ureteral cancer were 12.5%; in the general population, bladder cancer diagnoses out of total urothelial cancers are 89.9%, while those at ureteral level are 4.4% [33]. Furthermore, the fact that 6 out of 23 patients received a diagnosis of urothelial cancer before the diagnosis of IBD seems to shift the focus more onto the common risk factors of the two diseases, rather than on the action of drugs for IBD treatment. 

Our study has some limitations that warrant highlighting. Data loss bias is possible since a retrospective analysis was conducted. The long-time window may have led to non-homogeneous data collection over time. It should be noted, however, that all patients have been followed over the years by the same doctor (A.M.) in the same center, reducing the possibility of having dependent operator data. Another possible bias is linked to the presence of a smaller total population in the study compared to that of multicenter studies; ours, however, is the widest study with data relating to a single center. Finally, data about race, job, and family history (known risk factors of urothelial cancer) could have enriched the study but, due to the retrospective nature, were not available.

## 5. Conclusions

In conclusion, our study shows that patients with IBD have a similar risk of urothelial cancer compared to the general population. Nonetheless, smoking and being male seem to be risk factors associated with the development of this cancer, despite the fact that it is a single center study it should be taken into consideration. It would therefore be useful to adopt screening methods and prevention strategies for lifestyle changes in this specific population of male smokers with IBD. Finally, our study reassures the use of immunosuppressants in the treatment of IBD since it does not significantly increase the risk of developing urothelial cancer.

## Figures and Tables

**Figure 1 jcm-10-03257-f001:**
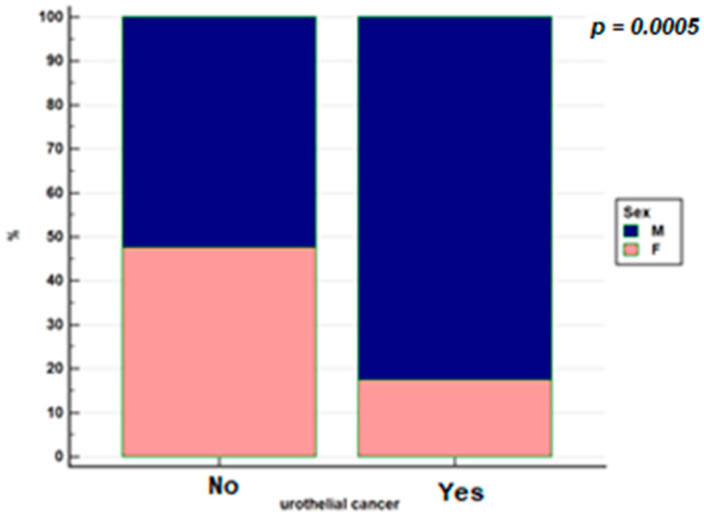
Distribution by gender in the population with IBD with urothelial cancer and without urothelial cancer.

**Figure 2 jcm-10-03257-f002:**
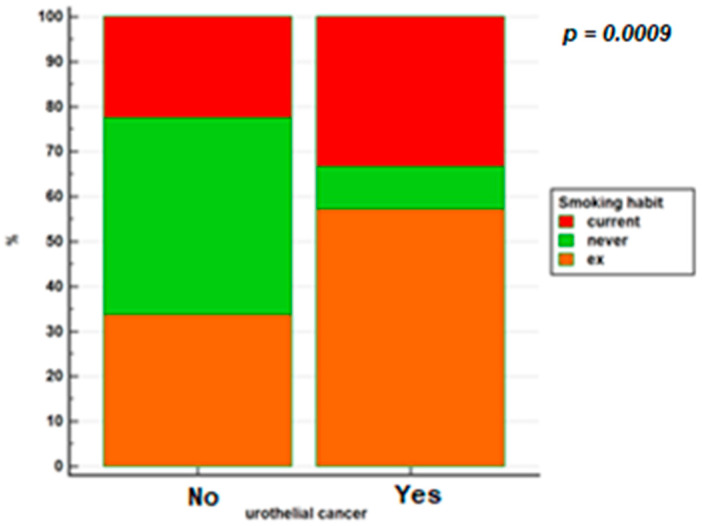
Smoking habits in patients with IBD based on whether or not they have developed urothelial cancer.

**Figure 3 jcm-10-03257-f003:**
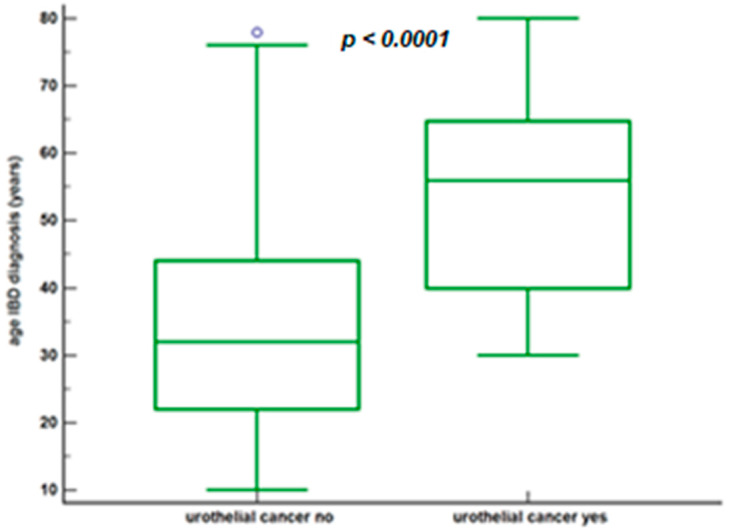
Age at diagnosis of IBD based on whether or not urothelial cancer has developed.

**Table 1 jcm-10-03257-t001:** Comparison of cases and controls.

Risk Factors	Cases (*n* = 23)	Controls (*n* = 250)	*p* Value
**Sex**			*p* = 0.005
Males	19 (82.6%)	131 (52.4%)	
Females	4 (17.4%)	119 (47.6%)	
**Smoking habit**			*p* = 0.009
Active smokers	7 (33%)	56 (22.5%)	
Former smokers	12 (57%)	84 (33.7%)	
Never smokers	2 (9.5%)	109 (43.8%)	
**IBD type**			*p* = 0.19
Crohn’s disease	17 (73.9%)	154 (61.1%)	
Ulcerative colitis	5 (21.7%)	93 (37.2%)	
IBD-U	1 (4.4%)	3 (1.2%)	
**Age at diagnosis of IBD**			*p* < 0.0001
Year (median, IQR)	56.0, 40.0–64.8	32.0, 22.0–44.0	
**Mesalazine**			*p* = 0.02
Yes	19 (82.6%)	237 (94.8%)	
No	4 (17.4%)	13 (5.2%)	
**Duration mesalazine**			*p* = 0.053
Months (median, IQR)	48.0, 8.25–188.25	88.5, 48.00–179.00	
**Thiopurine**			*p* = 0.38
Yes	5 (21.7%)	76 (30.4%)	
No	18 (78.3%)	174 (69.9%)	
**Duration thiopurine**			*p* = 0.79
Months (median, IQR)	40.00, 18.25–60.00	31, 6.00–82.00	
**Anti-TNF**			*p* = 0.012
Yes	1 (4.3%)	71 (28.4%)	
No	22 (95.7%)	179 (71.6%)	
**Surgical resection**			*p* = 0.69
Yes	8 (34.8%)	77 (30.8%)	
No	15 (65.2%)	173 (69.2%)	

IBD-U = inflammatory bowel disease unclassified; IQR = interquartile range; TNF = tumor necrosis factor.

## Data Availability

The data was collected anonymously. Anonymous data can be requested in case of need.

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
