# Peer review of "Risk Factors of Urothelial Cancer in Inflammatory Bowel Disease"

_jcm, 2021, doi:10.3390/jcm10153257_

Round 1

Reviewer 1 Report

This is a novel study and contains interesting findings. The manuscript is well-written. I have just one thing. Histopathology and grade of urotherial carcinoma can be added in the text.

Author Response

Dear reviewer, thank you for appreciating our study. From the point of view of tumor typing, the information present was partially incomplete and some histological information regarding the tumor was missing. According to the data in our possession, from the point of view of tumor grading, 9 (23.1%) tumors were in-filtrating (6 tumors were pTa, 3 were pT1), 1 (4.4%) tumor was infiltrating (pT3). From the point of view of tumor staging, 2 (8.7%) tumors were G1, 7 (30.4%) tumors were G2, 2 (8.7%) tumors were G3. We added these information in the text.

Reviewer 2 Report

The authors evaluated the risk of developing urothelial cancer in IBD patients because smoking is known as a risk factor for both diseases. They calculated the incidence rate of urothelial cancer in IBD and subsequently compared it with that of general population. The study found that urothelial cancer developed at the similar rate between the two groups and male and smoking were associated with the development of urothelial cancer in IBD.

Major comments:

  1. This study lacks a reasonable hypothesis to assess the possible association between IBD and urothelial cancer because sharing the same risk factor (i.e. smoking) does not implicate a causal relationship between the two. In fact, multivariate analysis found known risk factors of urothelial cancer.
  2. Race, job, and family history should be included in the analysis because they are known risk factors of urothelial cancer.
  3. The authors need to describe where the data of 250 controls comes from.

Author Response

Q1. This study lacks a reasonable hypothesis to assess the possible association between IBD and urothelial cancer because sharing the same risk factor (i.e. smoking) does not implicate a causal relationship between the two. In fact, multivariate analysis found known risk factors of urothelial cancer.

A1. Dear reviewer, thank you for your question.

In patients affected by IBD an association with extraintestinal malignancies (Pedersen N, Duricova D, Elkjaer M, et al. Risk of extraintestinal cancer in inflammatory bowel disease: metaanalysis of population-based cohort studies. Am J Gastroenterol 2010;105:1480-7.) has been found. Though mechanism underlying IBD and oncogenesis has not been clearly elucidated, it was reported that inflammation not only serves as host`s response to malignant tumor but also elicit carcinogenesis (Gakis G. The role of inflammation in bladder cancer. Adv Exp Med Biol 2014;816:183-9). Moreover, immunosuppressant medications, which are widely used for IBD treatment, are believed to be responsible for elevated risk of cancers like non-Hodgkin lymphoma, acute myeloid leukemia, non-melanoma skin cancers and urinary tract cancers (Bourrier A, Carrat F, Colombel JF, et al. Excess risk of urinary tract cancers in patients receiving thiopurines for inflammatory bowel disease: a prospective observational cohort study. Aliment Pharmacol Ther 2016;43:252-61). However, current studies about a possible relationship between IBD and urothelial cancer provided contrasting results.

The aim of our paper is to clarify, in a large cohort, if the risk of urothelial cancer is increased in IBD, eventually according to thiopurine use.

We added this rationale in the introduction, too.

Q2. Race, job, and family history should be included in the analysis because they are known risk factors of urothelial cancer.

A2. Dear Reviewer, we agree with you that these data could enrich the study but, due to the retrospective nature, they were not available. We added this to the criticisms of the study.

Q3. The authors need to describe where the data of 250 controls comes from.

A3. For each of the patients with identified urothelial cancer (cases), more than ten IBD patients who did not develop urothelial cancer (controls) were randomly selected (alphabetically) from the medical records of all patients with IBD followed in our gastroenterology clinic between 1978 and 2021 (we better specified it).

Reviewer 3 Report

Cristina et al., conducted a retrospective study to investigate an association between IBD and urothelial cancer. This manuscript is well written, and authors have accessed large IBD cohort. However, this manuscript lacks a conclusive finding to co-relate whether any IBD associated factor or patient’s past lifestyle habits has any effect on the prevalence of cancer. Other major concerns:

  1. Out of 5739 only 6 (0.1%) urothelial cancer patients preceded the diagnosis of IBD, still this is very limited to report significant, given the fact that almost 43 years of patient’s record analyzed. Authors have not included any in between medications prescribed to patients.
  2.  Table-1 It is not clear whether reported 6 patients were active or passive smokers.
  3. It seems a bias approach to take 1:10 non to cancer IBD patients in the control group.

Author Response

Q1. Cristina et al., conducted a retrospective study to investigate an association between IBD and urothelial cancer. This manuscript is well written, and authors have accessed large IBD cohort. However, this manuscript lacks a conclusive finding to co-relate whether any IBD associated factor or patient’s past lifestyle habits has any effect on the prevalence of cancer.

A1. Dear reviewer, thank you for appreciating our article.

In the literature it is debated whether urothelial tumors are more common in patients with IBD or whether the use of thiopurines is a risk factor for urothelial cancer. Our study concluded that urothelial cancers are no more prevalent in patients with IBD and the use of thiopurines is not a risk factor. This cancer is more common in patients with IBD if they are male or have a history of smoking, so this subgroup of patients with IBD needs special attention on this possible event.

Q2. Out of 5739 only 6 (0.1%) urothelial cancer patients preceded the diagnosis of IBD, still this is very limited to report significant, given the fact that almost 43 years of patient’s record analyzed. Authors have not included any in between medications prescribed to patients.

A2. Dear Reviewer, we agree with you that the absolute number of urothelial cancers is limited. On the other hand, ours is the single-center study with the largest data set. As regards the pharmacological history, given the retrospective nature of the study, only the data concerning the drugs taken to treat IBD were available: from the analysis carried out, both the use of mesalazine and the use of thiopurines proved to be safe. The latter data is of particular interest as the data in the literature in this regard are conflicting.

Q3. Table-1 It is not clear whether reported 6 patients were active or passive smokers.

A3. They were active smokers: we better specified it in the text.

Q4. It seems a bias approach to take 1:10 non to cancer IBD patients in the control group.

A4. All patients analysed came from the same cohort (patients with IBD followed in our gastroenterology clinic between 1978 and 2021). We compared patients with IBD who developed urothelial cancer versus patients with IBD who did not develop urothelial cancer to look for different risk factors. Since only a limited number of cases were found (23), a large number of controls (250) were selected to give sufficient statistical power to the study.

Round 2

Reviewer 2 Report

The authors amended the manuscript appropriately.

Author Response

Thank you very much for appreciating our effort to improve the manuscript according to your suggestions.

Reviewer 3 Report

Authors addressed earlier comments and incorporate substantial changes in the manuscript. Still the major concern with this study is the overall conclusion drawn that males with a history of smoking are at increased risk.

1. This is a very confined single center study and extrapolating the outcomes with overall national population is still questionable.

 2. IBD who developed urothelial cancer versus patients with IBD who did not develop, what is statistic with similar (23) in each group having similar age, sex and disease state?

3.  

Author Response

Authors addressed earlier comments and incorporate substantial changes in the manuscript. Still the major concern with this study is the overall conclusion drawn that males with a history of smoking are at increased risk. This is a very confined single center study and extrapolating the outcomes with overall national population is still questionable.

Dear Reviewer, thank you for appreciating our effort to improve our paper according to your suggestions.

Q1. This is a very confined single center study and extrapolating the outcomes with overall national population is still questionable.

A1. Dear Reviewer, the comparison with overall national population has been used to compare the frequency of urothelial cancer in IBD population versus general population. Risk factors for urothelial cancer were compared in a case-control setting comparing patients with IBD who developed urothelial cancer versus patients with IBD who did not develop urothelial cancer.

We softened the conclusions according to your suggestion.

Q2. IBD who developed urothelial cancer versus patients with IBD who did not develop, what is statistic with similar (23) in each group having similar age, sex and disease state?

A2. Dear Reviewer, controls (patients with IBD who did not develop urothelial cancer) were randomly chosen among the same IBD population. Data about age, sex, and disease state (use of thiopurine, anti-TNF) are reported in Table 1. We took into account the differences in the two groups by performing a multivariate analysis.